



# Marine heatwave characteristics in the Barents Sea: impact of changing baselines

Vidar S. Lien[1*], Roshin P. Raj[2], Sourav Chatterjee[3]

[1]Institute of Marine Research, Norway

[2]Nansen Environmental and Remote Sensing Center, Norway

[3]National Centre for Polar and Ocean Research, Ministry of Earth Sciences, India

[*]Correspondence to: Vidar S. Lien (vidar.lien@hi.no)

**Abstract.** Anomalously warm oceanic events, often termed marine heatwaves, can potentially impact the ecosystem in the affected region and has therefore become a hot topic for research in recent years. Determining the amplitudes and extent of marine heatwaves, however, depends on the definition and climatological baseline used. Moreover, the stress applied by the heatwave to the marine ecosystem will depend on which component of the ecosystem is considered. Here, we utilize a model reanalysis (1991-2021) to explore the frequency, intensity and duration of marine heatwaves in the Barents Sea, as well as their regional expression. We find that major marine heatwaves are rather coherent throughout the region and have comparable surface and bottom expressions. Moreover, we utilize a 60-year regional model hindcast to show the impact of changing baselines on marine heatwave statistics. Our results indicate that severe marine heatwaves are likely becoming more frequent in a future Barents Sea due to ongoing climate change.

## 1 Introduction

Marine heatwaves (MHW) are periods of warm spells in an ocean region, and are usually defined as periods when the temperature exceeds a given threshold based on a climatic baseline (e.g., Marbá et al., 2015; Hobday et al., 2016; Scannell et al., 2016). Due to the potential profound impact on marine life (e.g., Smale et al., 2019; Husson et al., 2022) and, hence, also socioeconomic impacts (Smith et al., 2021), MHW have received increasing attention in recent years, see Oliver et al. (2021) for a comprehensive review of recent literature. While the criteria to define MHW seem to converge to those proposed by Hobday et al. (2016), i.e., temperatures above the 90th percentile based on a fixed baseline, little attention has been given to the impact of the choice of baseline period, or climate normal, on the MHW characteristics and statistics such as frequency, intensity and duration (Chiswell, 2022). The underlying trends of global ocean warming (e.g., Cheng et al., 2022) and regional climate variability (e.g., Smedsrud et al., 2022) both impact the MHW statistics, and some regions may eventually enter a state of permanent MHW when compared with fixed baseline periods. As an example, while Fröhlicher et al. (2018) found a doubling of MHW days between 1982 and 2016 globally, Chiswell (2022) showed that accounting for climate change by removing the linear trend resulted in weaker MHW in the tropics and stronger MHW in the northern Pacific and Atlantic Oceans.



When MHW are calculated for a whole region, regional heterogeneities will be lacking, thereby reducing the applicability of
such an index. The Barents Sea is a complex shelf sea that mainly consists of a relatively warm and ice-free Atlantic Water
dominated part in the south, and a cold, seasonally ice-covered Arctic Water dominated part in the north. Moreover, both
regions have varying seasonal stratification (e.g., Smedsrud et al., 2013; Lind et al., 2018). The marine ecosystem is therefore
also differing between the two main regions, with further diversification within each region (see, e.g., Jakobsen and Ozhigin
(2011) for a comprehensive overview). However, the extension of the two regimes is changing due to ongoing climate change,
with the boreal, southern part expanding at the expense of the northern, Arctic part (e.g., Fossheim et al., 2015; Oziel et al.,
2020). The Barents Sea is home to several important, commercial fish stocks, in addition to a diverse marine ecosystem
including large groups of marine mammals and sea birds. Hence, MHW may have profound impacts on marine living
resources, especially with different species exhibiting differences in resilience to MHW events (e.g., Husson et al., 2022).
Here, we investigate the occurrences of surface and bottom MHW in four contrasting environments in the Barents Sea.
Moreover, we explore the differences in frequency, duration and intensity based on varying methodology for estimating MHW.
We also focus on the most severe MHW event in terms of cumulative degree-days and investigate its oceanic and atmospheric
preconditioning  and decay.
**2 Data & Methods**
**2.1 Model data**
We base our analysis on modeled daily averages from two different models; the EU Copernicus Marine Service ocean
reanalysis for the Arctic region based on the TOPAZ model system (Sakov et al., 2012; Xie et al., 2016; product ref 1, Table
1), hereinafter termed *TOPAZ reanalysis*. In addition, we have used a regional model hindcast utilizing the ROMS model
(Regional Ocean Modeling System; Shchepetkin and McWilliams, 2005) configured for the Nordic and Barents Seas region
(Lien et al., 2013, 2014, 2016; product ref 2, Table 1), hereinafter termed *ROMS regional hindcast*.
**Table 1: Products used and their documentation.**

| Product ref. no. | Product ID & type | Data access | Documentation |
|---|---|---|---|
| 1 | ARCTIC_MULTIYEAR_PHY_002_003; Numerical models | EU Copernicus Marine Service Product (2022) | Quality Information Document (QUID): Xie & Bertino (2022) Product User Manual (PUM): Hackett et al. (2022) |
| 2 | NordicSeas_4km, Numerical models | MET          Norway | Lien et al. (2013, 2014) |

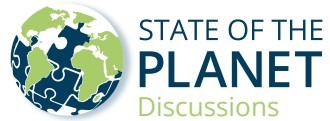

| | | Thredds Service | |
|---|---|---|---|
| 3 | Conductivity-Temperature-Depth data obtained in the Barents Sea | IMR database TINDOR (data accessible upon request) | |
| 4 | ERA5 Gridded Reanalysis (0.25 * 0.25 deg); monthly average on single level | EU Copernicus Climate Service Product (2023) | Hersbach et al., 2023 |


## 2.2 Ocean observation data

We have used available CTD (*Conductivity-Temperature-Depth*) casts (product ref. 3, Table 1) for assessing the performance
of the two model datasets with regard to bottom temperatures in four regions of the Barents Sea (Fig. 1). The CTD data were
obtained from the Institute of Marine Research database TINDOR (The Integrated Database for Ocean Research).

## 2.3 Atmospheric data

Monthly averages of turbulent heatfluxes and outgoing longwave radiation were downloaded from the EU Copernicus Climate
Service website (product ref. 4, Table 1).

## 2.4 Marine heatwave estimation method

We have adopted the definition of MHW proposed by Hobday et al. (2016), where MHW are defined as a period of more than
five days where the temperature is above the seasonally varying 90[th] percentile threshold relative to a predefined baseline
climatology of at least 30 years. Moreover, two consecutive events divided by a gap of two days or less are considered a single
event.
The TOPAZ reanalysis covers the time period 1991-2021. In compliance with common standards by the World Meteorological
Organization (WMO 2007; WMO 2015), we have chosen the period 1991-2020 as the climatological normal period. For the
ROMS regional hindcast, which covers the period 1960-2020, we have used two 30-year periods, 1961-1990 and 1991-2020.
These periods correspond to the previous and most recent, respectively, widely adopted climate normal periods. We choose
these periods to examine the effect on MHW statistics of using different baseline periods. The first period, 1961-1990, was a
relatively cold period in the Barents Sea region, whereas the period 1991-2020 has been relatively warm (e.g., González-Pola
et al., 2020).



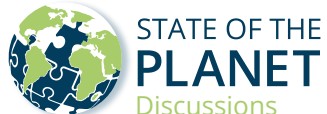

We have chosen four sub-regions where we compute the daily spatially averaged surface and bottom temperatures representing
contrasting marine environments: the Bear Island Trough in the south-western Atlantic Water inflow area to the Barents Sea;
the adjacent Spitsbergen Bank which represents a productive, shallow bank with an Arctic marine environment; the north-
eastern Barents Sea which represents the outflow region where strongly modified Atlantic-derived water masses leave the
Barents Sea; the Pechora Sea to the south-east which represents a shallow and coastal water influenced area (see map, Fig. 1).
Our Bear Island Trough region falls outside the full Barents Sea region, due to a compromise because of the orientation of the
grid, but it covers the area around 72°30'N where the core of the main inflow branch carrying Atlantic Water to the Barents
Sea is located (e.g., Skagseth et al., 2008).
For estimating MHW statistics we have used the python package provided by Eric C. J. Oliver:
https://github.com/ecjoliver/marineHeatWaves. Note, that the MHW detection algorithm counts every single MHW during a
year as a separate event, meaning that a single MHW event that extends over two or more calendar years can be counted several
times. This will impact the calculation of frequency and the associated trend in occurrences.

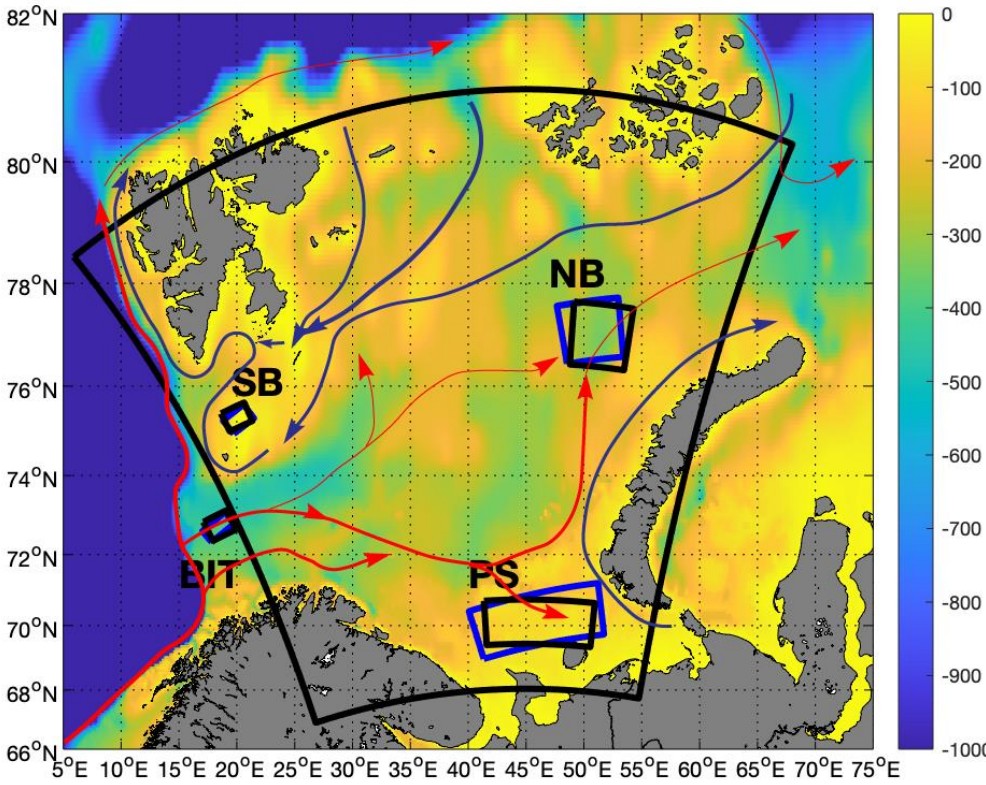






**Figure 1: Map of the Barents Sea. Colors show the bathymetry (in meters). Arrows show the main current patterns for Atlantic Water (red) and Arctic Water (blue). Boxes show regions for estimating marine heatwaves statistics from the TOPAZ reanalysis (black) and ROMS regional hindcast (blue). BIT: Bear Island Trough; NB: north-eastern Barents Sea; SB: Spitsbergen Bank; PS: Pechora Sea.**

## 2.4 Model evaluation

Even though both the model products used in this study have been evaluated in previous studies (for TOPAZ reanalysis, see: Lien et al. (2016); Xie et al. (2019, 2023); for ROMS regional hindcast see: Lien et al. (2013, 2014, 2016)), we provide direct comparison with observations of near-bottom temperature from CTD casts where available in the four sub-regions. The motivation for comparing only bottom temperatures is that satellite sea surface temperature observations are assimilated into the TOPAZ reanalysis. Moreover, the sea surface temperature is constrained by ocean-atmosphere bulk fluxes. Furthermore, the novelty of our study is the analysis of MHW events near the ocean bottom, in comparison to previous studies focusing only on MHW events at the sea surface (e.g., Mohamed et al., 2022).

Here, we compare modelled and observed near-bottom temperatures averaged in time (monthly) and space (see sub-regions, Fig. 1). The modelled seasonal signal was removed from both model and observation timeseries before the correlation was calculated. The comparisons are summarized in Table 1 and Supplementary Figure S1.

**Table 2: Statistics summarizing the comparison between the models and observations. Correlations are shown in boldface when $p < 0.05$ and underlined boldface when $p < 0.01$. BIT: Bear Island Trough; SB: Spitsbergen Bank; PS: Pechora Sea; NEBS: North-Eastern Barents Sea.**

| Model | Statistic | BIT | SB | PS | NEBS |
|---|---|---|---|---|---|
| TOPAZ | $N$ | 202 | 49 | 34 | 11 |
| | Bias [°C] | 1.9 | -2.1 | -0.8 | -0.6 |
| | RMSd [°C] | 2.0 | 2.4 | 1.0 | 0.7 |
| | Correlation [$r$] | **0.55** | **0.39** | **0.78** | **0.66** |
| ROMS | $N$ | 237 | 59 | 41 | 16 |
| | Bias [°C] | 0.2 | 0.5 | -0.0 | -0.5 |



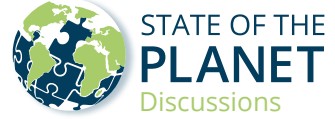

| | RMSd [°C] | 1.0 | 1.2 | 0.8 | 0.5 |
|---|---|---|---|---|---|
| | Correlation [$r$] | **0.36** | 0.25 | **0.64** | **0.63** |


## 3 Results

We first estimate MHW statistics based on the TOPAZ reanalysis for the full Barents Sea region (see Fig. 1 for area definition).
Two distinct MHW events are identified in both the surface and bottom temperature time series. While the strongest MHW,
in terms of cumulative effect (degree days), appeared in 2016 both near the surface and near the bottom, the second strongest
MHW appeared in 2013 near the surface and in 2012 near the bottom (see Supplementary Figures S2 and S3 for the full
timeseries). When applying the MHW definition provided by Hobday et al. (2016), we computed the following MHW statistics
(summarized in figure 2). Near the surface, the 2016 MHW had an average intensity of 1.25 °C above the climatological
temperature and a total duration of 512 days (from November 18, 2015, to April 12, 2017). The near-bottom expression had
an average intensity of 1.02 °C above the climatological temperature and a total duration of 587 days (December 30, 2015, to
August 7, 2017). However, while the surface and bottom expression of the 2016 MHW are comparable, we find some
differences between the average surface and bottom expressions of MHW in the Barents Sea. The frequency of MHW for the
1991-2021 period was found to be 0.61 year$^{-1}$ near the surface and 0.23 year$^{-1}$ near the bottom, while the average maximum
intensity was found to be 1.33 °C and 0.92 °C near the surface and bottom, respectively. The duration was, on average, longer
near the bottom (183 days) compared with near the surface (45 days). The frequency and maximum intensity had positive
trends both near the surface and near the bottom, while the duration had a positive trend near the surface and a negative trend
near the bottom. However, these statistics need to be interpreted with care. For example, while we identified two main MHW
events, several shorter periods were also classified as MHW, especially within the near-bottom (Supplementary Figures S4a,
S5a). Some of these events were related to the same warm period but with intermittent periods with temperatures below the
90th percentile in between. These shorter periods affected the calculation of the duration trend. Thus, although all the near-
bottom MHW events detected occurred within the last 18 years of the 1991-2021 period, the average duration appeared with
a negative trend in the calculations.



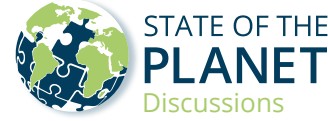



**Figure 2: Number of marine heatwave events per year (top), maximum intensity of the heatwave events (middle) and average marine heatwave duration (bottom) for the full Barents Sea and four sub-regions during the period 1991-2021. The associated decadal trends are shown in hatched colors. The trend is provided in boldface if significant to 95% ($p < 0.05$). Surface values are shown by blue bars and bottom values are shown by red bars. Based on data from the TOPAZ reanalysis.**

To look for regional differences, we investigate the 2016 MHW event in the four sub-regions depicted in figure 1. In the TOPAZ reanalysis, the 2016 event was the most severe MHW during the full 1991-2021 period in three out of the four sub-regions investigated: the Northeast Basin, the Spitsbergen Bank, and the Pechora Sea. In the Bear Island Trough, the temperature anomaly classified as an MHW intermittently throughout 2016 and while the cumulative impact in terms of degree days was largest in 2016 (at the surface; Supp. Fig. S4b), 2012 experienced the most severe continuous MHW (Fig. 3, 4). Other regional differences include that in the downstream north-eastern Barents Sea, the surface expression of the 2016 MHW was most severe in the first half of 2016, while on the Spitsbergen Bank it was most pronounced in the second half of 2016 (Fig. 4). In the Pechora Sea, the 2016 MHW persisted throughout the whole year. Moreover, the intensity of the MHW increased downstream in the Barents Sea, from an average 1.21 °C and 0.60 °C above the climatology near the surface and bottom, respectively, during the most severe part of the 2016 MHW event in the Bear Island Trough, to 1.54 °C and 1.68 °C, respectively, in the north-eastern Barents Sea and 2.45 °C and 1.51 °C, respectively, in the Pechora Sea. On the Spitsbergen Bank, the average intensity was 2.28 °C near the surface and 2.25 °C near the bottom.

In all the sub-regions, except for the Spitsbergen Bank where the water column is well-mixed due to tidal mixing, the MHW frequency is larger near the surface than near the bottom (Fig. 2a). In the Bear Island Trench and the Pechora Sea, the maximum intensities of the MHW near the surface are approximately twice as large as the maximum intensities near the bottom, whereas in the north-eastern Barents Sea and on the Spitsbergen Bank the intensities are similar near the surface and bottom (Fig. 2b). However, the MHW near the bottom tend to be more persistent, as seen from the longer average duration (again, the Spitsbergen Bank is an exception; Fig. 2c).







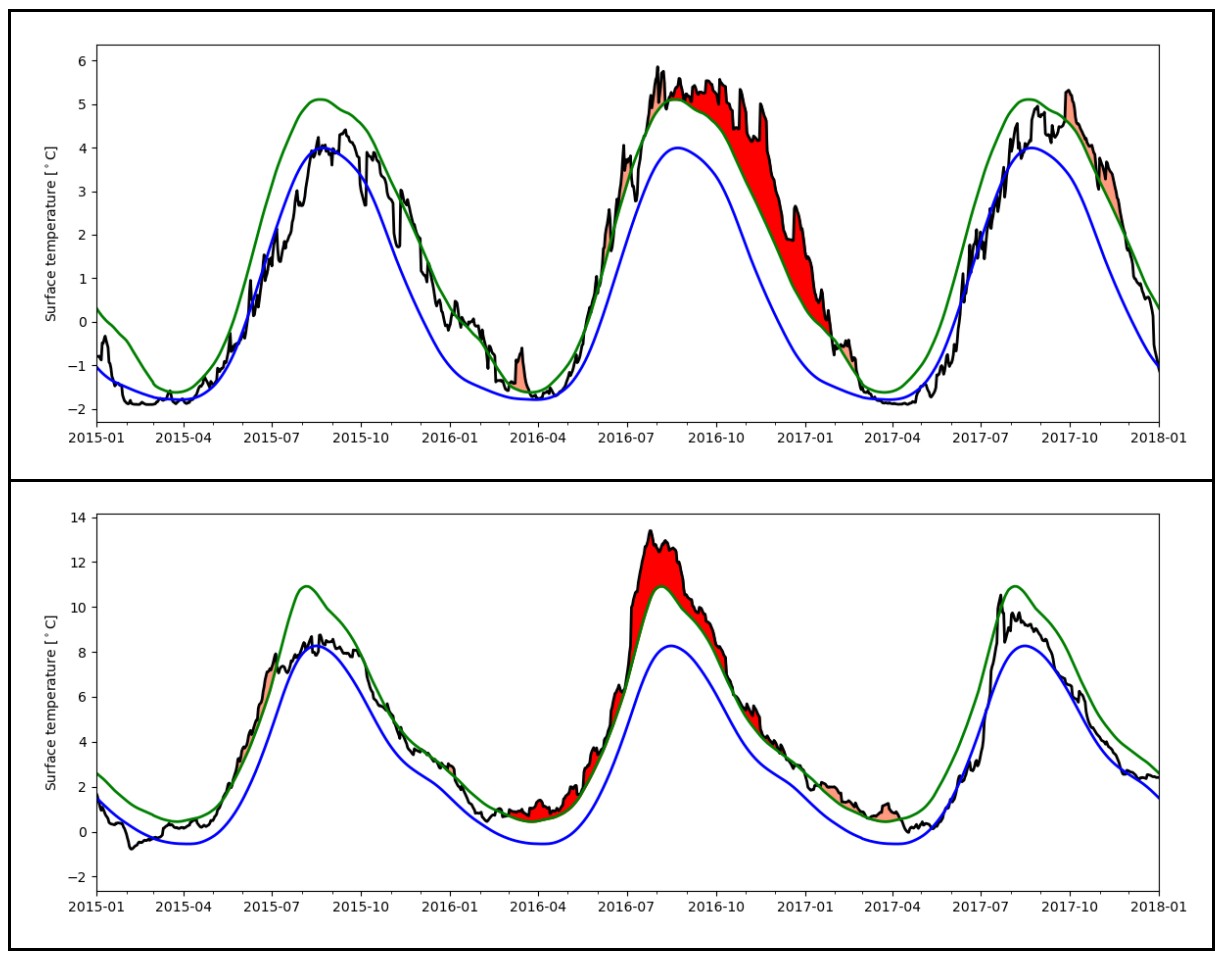

157

**Figure 3: (Top) Time series (black lines) showing the temperature at 5-meter depth spatially averaged over the Barents Sea. Blue lines show daily climatology. Green lines show the upper 90th percentile. The most intense marine heatwave in terms of cumulative degree days for the full 1991-2021 period is shown in dark red shading. Other marine heatwaves are shown in pink shading. Subpanels show the following sub-regions (from top to bottom): The Bear Island Trough, The north-eastern Barents Sea, The Spitsbergen Bank, The Pechora Sea. All panels show the period January 1st 2015 to January 1st 2018. Note the different scales on the y-axes.**

164







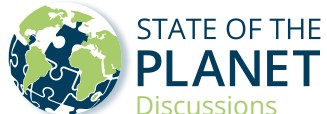

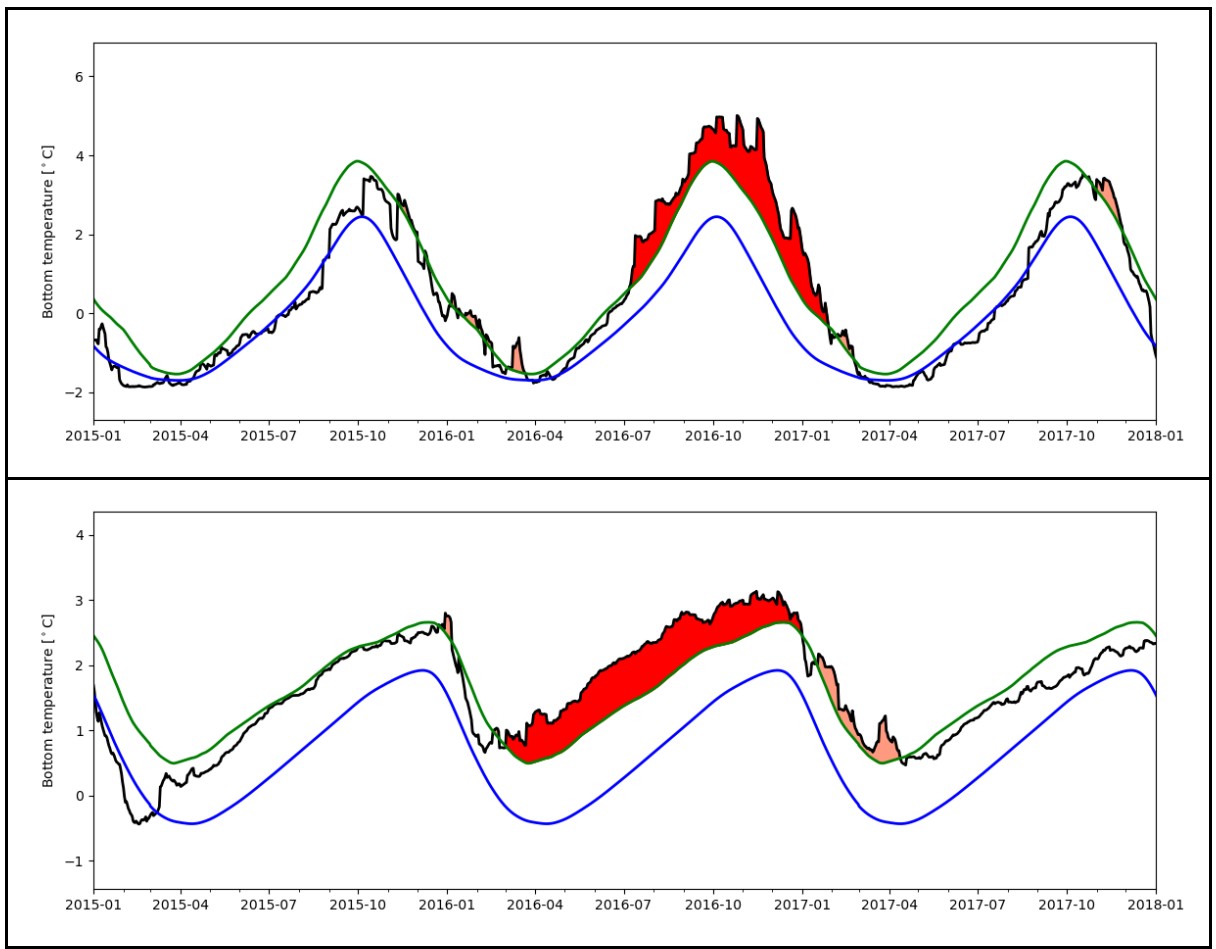

**Figure 4: Same as Figure 3, but showing near-bottom temperatures.**

### 3.1 Preconditioning and atmospheric forcing of 2016 MHW event

Leading up to the onset of the 2016 MHW, the inflow of warm Atlantic Water to the Barents Sea was above average during the whole of 2015 (ICES, 2022). During the subsequent winter of 2015/16, the turbulent heat loss however (between 20 and 70 W/m2) was below the climatological average (1993-2021) in the southern Barents Sea (i.e., along the Atlantic Water pathway through the Barents Sea) despite the increased advection of oceanic heat (Fig. 5a,e). Furthermore, for the analysis period (1993-2021), the area-averaged turbulent heat loss in the southern Barents Sea (25-45E; 71-75N) was also the lowest during the onset of the 2016 MHW event (not shown). Thus, an increased Atlantic Water heat transport and reduced heat loss to the atmosphere resulted in the development of this strong MHW event during 2016. In the following winter of 2016/17, i.e., during the decay of the 2016 MHW event, turbulent heat loss and outgoing longwave radiation in the northern Barents Sea (25:45E; 76-80N; Fig. 5b,e,f) reached the largest values, possibly due to record low winter sea ice extent and negative cloud



cover in the northern Barents Sea (not shown). However, in the southern Barents Sea no obvious changes in heat loss from the
ocean surface is observed (Fig. 5b). Rather, the decay of the MHW event in the southern Barents Sea appears to be caused by
the decrease in Atlantic Water transport across the Barents Sea Opening during 2016 (ICES, 2022). Thus, the onset and decay
of the 2016 MHW event in the Barents Sea can be linked to the combined influence of Atlantic water transport into the Barents
Sea and oceanic heat loss in the southern and northern Barents Sea.



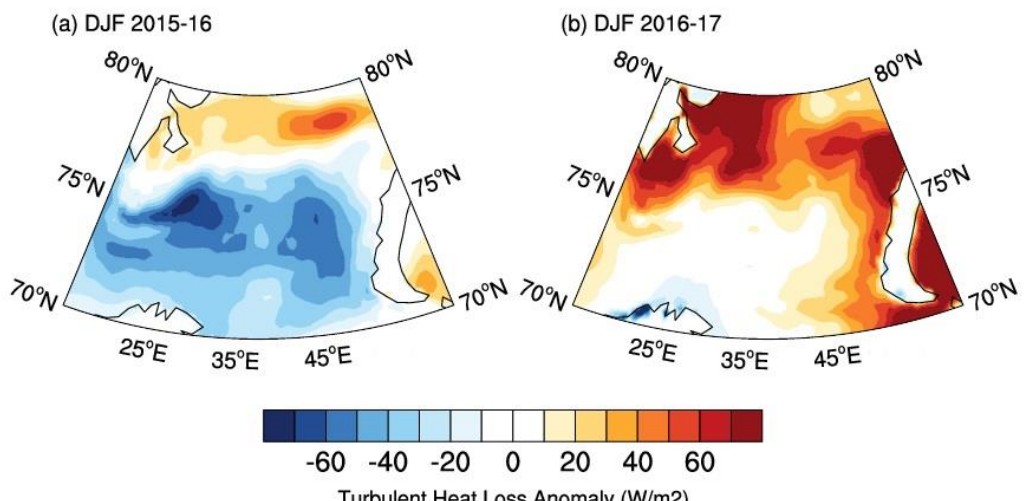

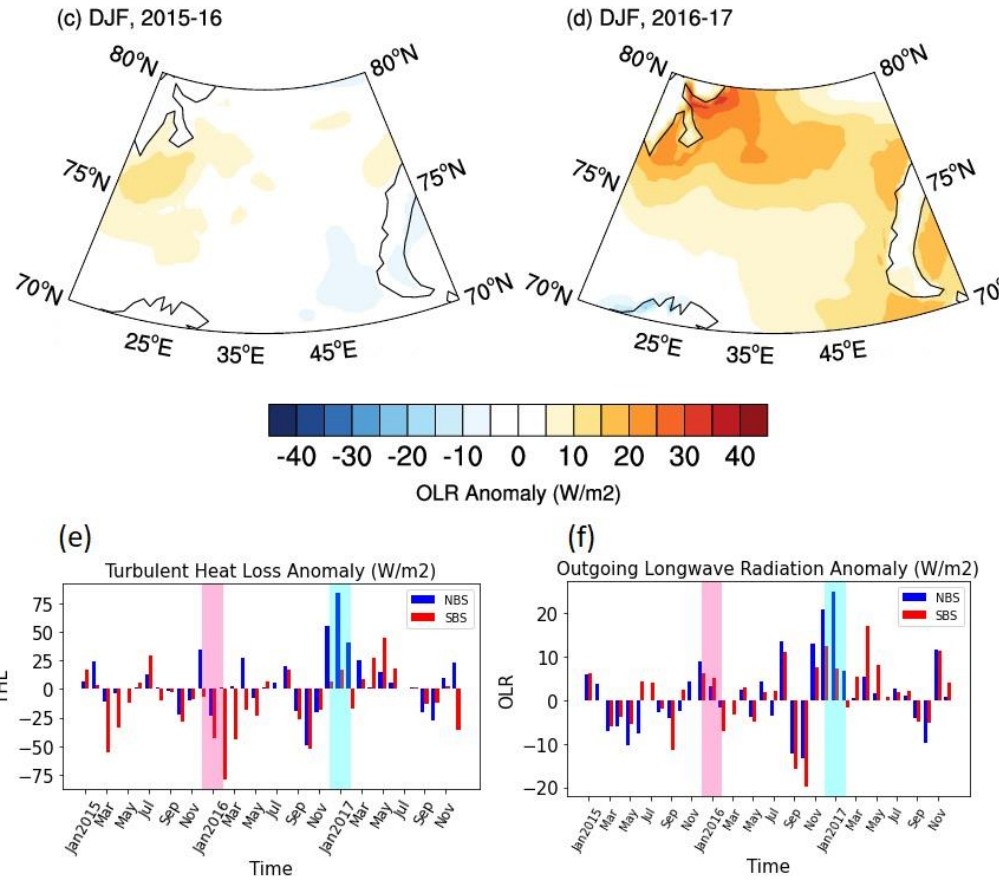



**Figure 5: Atmospheric preconditioning leading up to the MHW depicted in Fig. 2. (a,b) DJF (December(-1), January, February (0)) turbulent (latent + sensible) heat loss anomaly (W/m2) for 2016 (a) and 2017 (b). Same as (a,b) but for Outgoing Longwave Radiation (OLR). Positive values indicate upward fluxes. Monthly mean turbulent heat loss (e) and OLR (f) over northern (blue, 25:45E; 76-80N) and southern (red, 25-45E; 71-75N) Barents Sea. The onset (DJF, 2015-16) and decay (DJF, 2016-17) phase of the 2016 MHW event are shaded in pink and cyan colours. Data: ERA5**

### 3.2 Effect of changing baselines

Next we investigate the effect of changing the baseline when calculating the MHW statistics, using the results from the ROMS regional hindcast. Common for all the four sub-regions is that the frequency of MHW occurrences decreased by approximately one half (two thirds in the north-eastern Barents Sea) when changing the baseline from the 1961-1990 climate normal to the 1991-2020 climate normal (Table 3). On the other hand, the decadal trend in the number of occurrences not only increased, but are also statistically significant to the 95% level ($p < 0.05$) in all regions. Oppositely to the frequency, the average maximum intensity was comparable when using the two different baselines in most of the sub-regions (Table 4). As an exception, in the Pechora Sea, the average intensity increased, especially near the surface, when using the 1991-2020 climate normal as baseline. The explanation for the increase in maximum intensity when comparing with a higher climatological average temperature is that several weaker warm events were no longer classified as MHW (not shown).

On average, the MHW duration decreases when changing the baseline from the 1961-1990 climate normal to the 1991-2020 climate normal, although the differences are small in the Bear Island Trough and on the Spitsbergen Bank (both in the western Barents Sea; Table 5). A striking difference between the two western sub-regions and the two eastern sub-regions is the transition from more well-mixed condition to more of a stratified two-layer system, causing a decoupling between the surface and bottom conditions. As a result, near-bottom MHW show considerably longer duration than surface MHW in the north-eastern Barents Sea and in the Pechora Sea, whereas in the Bear Island Trough and on the Spitsbergen Bank the duration near the surface and near the bottom are comparable. Indeed, when using 1961-1990 as the baseline in the north-eastern Barents Sea, the most severe MHW in terms of cumulative degree days appears at the end of the 60-year period with the last five years (starting March 10, 2015) of the timeseries representing an MHW (not shown). Thus, the area has entered a state of permanent MHW when choosing this older baseline, which explains the strongly positive trend in duration (129 days per decade; Table 5). Moreover, several MHW events appear throughout the 60-year period, including one MHW event at the start of the timeseries in 1961 (not shown). When the baseline is changed to the 1991-2020 period, two distinct MHW appear, in 2016 and 2018, with the 2016 event being the most severe and no MHW event is detected prior to 2007.





**Table 3: Number of marine heatwave events per year during the period 1961-2020 +/- the decadal trend for two different baseline periods, 1961-1990 and 1991-2020. The baseline period 1991-2020 is also used for the detrended, full time series (1961-2020). The trend is provided in boldface if significant to 95% (*p* < 0.05), or in italics if not significant (*p* > 0.05). Values for the surface are shown on top and values for bottom are shown below. BIT: Bear Island Trough; SB: Spitsbergen Bank; PS: Pechora Sea; NEBS: North-Eastern Barents Sea.**

| Baseline \ Area | BIT | SB | PS | NEBS |
|---|---|---|---|---|
| 1961-1990 | **2.30** + *0.29* <br> **2.15** + *0.06* | **1.73** + *0.20* <br> **1.70** + *0.22* | **1.38** + *0.16* <br> **0.92** + *0.13* | **1.85** + *0.29* <br> **0.63** + *0.11* |
| 1991-2020 | **0.95** + **0.31** <br> **0.90** + **0.28** | **0.82** + **0.21** <br> **0.80** + **0.19** | **0.57** + **0.22** <br> **0.52** + **0.22** | **0.63** + **0.33** <br> **0.22** + **0.19** |

**Table 4: Same as Table 4, but showing average maximum intensity (in °C).**

| Reference period \ Area | BIT | SB | PS | NEBS |
|---|---|---|---|---|
| 1961-1990 | **1.32** + *0.05* <br> **1.38** + *0.04* | **1.20** + *0.04* <br> **1.22** + *0.03* | **1.73** + **0.20** <br> **1.31** + **0.07** | **1.08** + **0.16** <br> **0.62** + **0.09** |
| 1991-2020 | **1.39** + *0.02* <br> **1.56** + *0.05* | **1.26** + **0.10** <br> **1.26** + **0.10** | **2.30** + **0.08** <br> **1.54** + **0.05** | **1.47** - **0.28** <br> **0.71** - *0.17* |

**Table 5: Same as Table 4, but showing average duration (in days).**

| Baseline \ Area | BIT | SB | PS | NEBS |
|---|---|---|---|---|
| 1961-1990 | **25.2** + **5.2** <br> **30.3** + **7.6** | **32.1** + **6.6** <br> **32.7** + **5.4** | **50.5** + **16.7** <br> **72.9** + *15.3* | **63.0** + **22.6** <br> **284** + *129* |
| 1991-2020 | **18.0** + **3.1** <br> **20.2** + *5.1* | **28.4** + **7.6** <br> **28.6** + **7.8** | **32.8** + **10.0** <br> **55.1** + *3.1* | **33.3** - *0.1* <br> **99.2** - *28.9* |

**4 Discussion**

We have estimated MHW frequency, duration and intensity near the surface and the bottom in the Barents Sea, based on an ocean reanalysis for the period 1991-2021. Moreover, we have investigated the impact of changing baselines when estimating MHW statistics in the Barents Sea, based on a regional hindcast for the period 1961-2020. We find that the Barents Sea generally experiences few, but pervasive MHW that affect the whole region.



Previous studies of MHW, including the Barents Sea, have mainly focused on the ocean surface due to the availability of
satellite remote sensing sea surface temperature data (e.g., Mohamed et al., 2022). Our results indicate significant MHW events
also near the ocean bottom, exemplified by MHW events in part related to changes in sea-ice conditions, and the bottom
expressions of the MHW tend to last longer. We have shown that, in the north-eastern Barents Sea, the ocean bottom layer
appears to have entered a state of permanent MHW when using a 1961-1990 baseline. Moreover, the average duration of
bottom MHW are approximately three times longer or more than for surface MHW in this sub-region, independent of the
choice of baseline. The explanation for the strong MHW signal near the bottom in this area is likely the strong reduction in
sea-ice formation on nearby banks. This area is one of the regions that has experienced the largest changes in the sea-ice cover
in recent decades (e.g., Yang et al., 2016; Onarheim and Årthun, 2017) and has thus experienced a strong reduction in the
formation of cold, brine-enriched bottom water sinking into the deeper parts of the north-eastern Barents Sea (Midttun, 1985;
Lien & Trofimov, 2013). Occasional presence of such cold bottom water further west in the Barents Sea has been hypothesized
to cause differences in the position of the Polar Front as detected by bottom living organisms compared with hydrographic
properties in the pelagic zone (Jørgensen et al., 2015). Thus, the transition indicated by bottom MHW in the north-eastern
Barents Sea may have a profound impact on bottom fauna by allowing boreal species with less resilience to below-zero
temperatures to settle.
Previous findings by Mohamed et al (2022), based on satellite remote sensing sea-surface temperature data, contrasted the
Spitsbergen Bank area showing no trend in MHW frequency and cumulative duration with the Pechora Sea area showing
significant trends in both frequency and duration. None of the two regions showed significant trends in MHW mean intensity.
Our findings agree with those of Mohamed et al. (2022) that the Pechora Sea has experienced a positive trend in both MHW
frequency and duration at the surface. However, our results indicate that there is no significant trend in MHW duration near
the bottom. Moreover, our results do show positive trends in both the MHW frequency and duration on the Spitsbergen Bank,
although we did not find a statistically significant trend in MHW intensity on the Spitsbergen Bank. However, the Spitsbergen
Bank is also the area where the TOPAZ reanalysis shows the largest bias and RMS deviation, as well as the lowest correlation
when compared with in-situ temperature observations. Thus, we cannot draw firm conclusions whether our results for the
Spitsbergen Bank area contradict the findings of Mohamed et al. (2022).
Our findings that the strong 2016 MHW event was preceded by stronger than average Atlantic Water inflow and anomalously
weaker ocean-to-atmosphere heat loss further suggest that MHW may become more frequent and severe in terms of intensity
and duration in a future Barents Sea with continued increase in oceanic heat advection from the North Atlantic (e.g., Årthun
et al., 2019) in combination with reduced ocean-to-atmosphere heat loss within the Barents Sea (e.g., Skagseth et al., 2020).




## 5 Data availability

A list of the data products utilized in this paper, along with their availability and links to their documentation, is provided in Table 1.

## 6 Author contribution

All authors contributed to the design, analysis, and writing of the paper.

## 7 Competing interests

The authors declare that they have no conflict of interest.

## 8 Acknowledgements

This work was funded by the Copernicus Marine Service, contract #21002L1-COP-MFC ARC-5100.

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
