# Peer review of "Surface and bottom marine heatwave characteristics in the Barents Sea: a model"

_State of the Planet, 2023_

## Referee Comment (RC2)

This is a study of the characteristics of marine heatwaves in the Barents Sea. It aims to investigate the occurrence of surface and bottom heatwaves in four different environments in the Barents Sea, to explore the differences in marine heatwaves characteristics when changing baselines, and to focus on the most severe MHW events in their study area.

Overall, I vacillated between rejecting the paper and accepting it with major comments and decided to give the authors another chance to improve their work. I have major and minor comments, such as the following:

**Major comments:**

1- The language used to describe the results of this work is very general and has nothing to do with the MHW topic. To be more clear, there are some terminologies that are known to be used by researchers of different scientific topics, here in this paper they used different terminologies that are far from what should be used in a study like the MHWs, and this led to a lot of confusion when I read their paper. I doubt that the authors used some sort of paraphrasing software that caused these repeated mistakes in their manuscript. Here are examples of the mistakes to which I refer:

- **Line 9:** It was written "Determining the amplitudes and extent of marine heatwaves". For MHWs, the term "**intensity**" was used to describe the level that the SST reached above the threshold climatology, while the term "amplitude" is usually used for ocean waves, sound waves, etc., not for MHWs. Also, the extent of the MHWs could be a vertical or spatial extent, so please specify which one you are referring to, and I think you mean vertical extent since you don't have any spatial maps in your paper.

- **Lines 13 – 14:** The authors used the word "expression" in two different places and this word does not mean the same thing in the 2 places. The first time it said "to explore the frequency, intensity, and duration of marine heatwaves in the Barents Sea, as well as 13 their regional expression", and I suspect they were referring to the **regional occurrence** of MHWs. The second place was, "We find that major marine heatwaves are rather coherent throughout the region and have comparable surface and bottom expressions", and I believe using words like **occurrences or events** is more representative in this case. Please change it through the entire manuscript.

- **Line 23:** "temperatures above the 90th percentile based on a fixed baseline", Hobday et.al., 2016 define an MHW when the SST exceeds either **the $90^{th}$ percentile of the moving climatological average** or **a fixed threshold**, so the above statement is not correct, please correct it. Also superscript the "th" that is after 90.

- **Line 24:** "Climate normal", it is usually referred to as climatological average or climatological mean, please change it.

- **Line 32:** "When MHW are calculated for a whole region, regional heterogeneities will be lacking, thereby reducing the applicability of such an index" ??? which index???????

2- **In the introduction:** you refer to other studies in the literature (e.g., line 37 (see e.g., Jakobsen and Ozhigin (2011) for a comprehensive overview)). Instead, try to make your introduction richer and like a story that gives the reader an idea of the topic you are studying. There are so many studies on MHWs and one of them deals with the Barents Sea (https://doi.org/10.3389/fmars.2022.821646). So you can use their work to introduce yours, and the authors should also mention how their work differs from the previous one. Also, you have a very unique area of study. I would have expected you to describe it in more detail in your introduction. This will help the reader to know more about the circulation in the Barents Sea, the physical properties and the atmospheric conditions, so that later when you talk about it, they can compare the normal and the extreme conditions.

3- **Lines 70 - 73:** "We choose these periods to examine the effect on MHW statistics of using different baseline periods. The first period, 1961-1990, was a relatively cold period in the Barents Sea region, whereas the period 1991-2020 has been relatively warm (e.g., González-Pola et al., 2020)". I don't really see any logic in comparing the MHWs defined by these two baselines, because the result is easy to expect. It was a cold period from 1961 to 1990, so this baseline will result in longer and more intense MHWs compared to the warm period baseline (1991-2020). If the author wants to study the results of MHWs using different baselines, I suggest you try to take a longer or shorter period for climatology, e.g. 25 or 35 years, but within the same period (warm or cold) so that you can check whether using less or more than 30 years of climatology could be reflected in the characteristics of MHWs or not. This will be very helpful for the studies examining the composite events of the MHWs and, for example, low chlorophyll-a events, since the available data for chlorophyll-a start from 1998.

4- **Lines 101-102:** "Here, we compare modelled and observed near-bottom temperatures averaged in time (monthly) and space (see sub-regions, Fig. 1)." How did you manage to define the MHWs using the monthly data? According to the definition you use, it is a daily phenomenon.

5- **Lines 111 – 114:** "We first estimate MHW statistics based on the TOPAZ reanalysis for the full Barents Sea region (see Fig. 1 for area definition). Two distinct MHW events are identified in both the surface and bottom temperature time series. While the strongest MHW, in terms of cumulative effect (degree days), appeared in 2016 both near the surface and near the bottom, the second strongest MHW appeared in 2013 near the surface and in 2012 near the bottom".

- Does it make sense that you find only 2 MHW events over 31 years ( 1991 – 2021)? Especially the other study (Mohammed et.al., 2022) stated that over the 39 years (1982-2020) In total, 72 MHWs were recorded in the whole Barents Sea over the 39-years study period with a total of 1,068 MHW days.

- How is it possible to find MHWs on the bottom (2012) that cannot be seen on the surface? Especially your area is very shallow (maximum depth about 400 m), where does the heat come from?

6- **Line 115:** "When applying the MHW definition provided by Hobday et al. (2016)" . So what was the method used to define MHWs in lines 111 – 114?

7- I would like to ask you if you have studied the role of the sea ice cover cycle in the region for the occurrence of MHWs?

**Minor comments:**

- In your data section please specify the periods that you used for each dataset
- Lines 151 - 152 rephrase it to be "In the Bear Island Trench and Pechora Sea, the maximum intensities of MHW near the surface are about twice the maximum intensities near the bottom".
- The figures: Put labels (A, B, ….etc) on the panels so it easier to be easier to be interpreted.

---

## Author Response (AR1)

**Reply to Reviewer #1**

The authors presented a study of MHWs in the Barents Sea, which is very interesting since the MHWs were less studied in the high latitudes. However, the study cannot be published at the current form for the following major and minor comments:

**Reply:** We would like to thank the reviewer for all the constructive comments, which we think helped improve the manuscript.

Also, please note that, in addition to addressing the issues raised by this reviewer, we have also updated all results, figures, and tables to also include the year 2022, because that year was not available at the time of the submission of the manuscript.

**Major comments:**

(a) surface and bottom MHWs

It is very interesting to see the bottom MHWs in this study. My impression is that it might be better to have their focus on the comparisons of surface and bottom MHWs, and even change the title of the manuscript.

However, it is not clear whether the MHWs in the section using ROMS data are for the surface or bottom. It might be good to compare both surface and bottom MHWs just as using the TOPAZ data,

**Reply:** Because we have reworked the analysis on changing baselines using changing climatic window length within the timeframe of the TOPAZ data (1991-2022) only, we have removed the reference to ROMS data altogether.

Moreover, we have changed the title of the manuscript, as suggested by the reviewer, as follows:
***Surface and bottom marine heatwave characteristics in the Barents Sea: a model study***

(b) Baseline comparison

I don't think this is interesting, since the results are very intuitive at least qualitatively.

I don't think this is meaningful either, since this is just a way for scientists to redefine MHWs for a later period of baseline, but the ecosystem may not be able to get used to the new baseline quickly unless the authors can provide the physical evidence.

**Reply:** While we agree that the outcomes of the comparison of baselines are intuitive, the comparison was motivated by the different response times of the different components of the marine ecosystem. However, we have performed new analysis using different baselines with less intuitive outcomes, as also suggested by reviewer #2, by using changing climatic window length (1991-2020; 1996-2020; 2001-2020).

(c) MHW algorithm

It is not clear whether the MHWs were diagnosed from the entire time series from 1991 to 2021, or rather diagnosed year by year from January 1 to December 31, since the statement in L83-84 is not consistent with that in Oliver et al (2018). See my detailed comments for L83-84.

**Reply:** We agree that this statement was confusing and have removed it and rather clarified by stating that we followed the procedures as outlined by Hobday et al. (2016) using the algorithm as provided by Oliver et al. (2018).

(d) Heat budget analysis
I am glad to see the heat budgets were used to explain the changes in MHWs, but budgets should be closed. The increase of influx from the Atlantic may not be necessary in favor to MHWs, if the outflux is considered. Also, it is not clear whether theses heat fluxes can be used to explain both surface and bottom MHWs.
**Reply:** Thank you for pointing out that this part which was not clear enough. We have added the following sentence with reference to earlier studies on the relation between heat advection into the Barents Sea and the heat content within the Barents Sea, as well as the finding that the heat transported out from the Barents Sea is generally an order of magnitude less than the heat transported into the Barents Sea:
*While we did not perform a closed heat budget calculation, we note that the oceanic heat carried by the downstream outflow from the Barents Sea has previously been reported to be smaller than the inflow by an order of magnitude (Gammelsrød et al., 2009; Smedsrud et al., 2013). Furthermore, a previous study found that increased heat advection to the Barents Sea lead to increased ocean heat content in the interior Barents Sea (Lien et al., 2017).*

**Minor comments:**
L13-14, It is not clear for "surface and bottom expressions"
**Reply:** We have rephrased the text to clearly state that our analysis shows that the characteristics of the major MHWs are comparable near the sea surface and the sea floor.

L18-19, the recent studies on the MHWs in the Arctic (Hu et al. 2020 and Huang et al. 2021) are worth citing here.
Hu, S., Zhang, L., & Qian, S. (2020). Marine heatwaves in the Arctic region: Variation in different ice covers. Geophysical Research Letters, 47, e2020GL089329. https://doi.org/10.1029/2020GL089329
Huang, B., Z. Wang, X. Yin, A. Arguez, G. Graham, C. Liu, T. Smith, H.-M. Zhang, 2021: Prolonged Marine Heatwaves in the Arctic: 1982-2020. Geophys. Res. Lett., 48, e2021GL095590, https://doi.org/10.1029/2021GL095590.
**Reply:** Thank you for the recommendations. We have included these relevant references.

L23, I always have difficulty to understand why 90[th] percentile was selected as an MHW criterion, since from statistics point of view the 90[th] percentile is really too low.
**Reply:** We chose the 90[th] percentile based on previous published literature, and also because our study is investigating MHWs in general. However, we do agree that in more specialized studies other criteria may be more appropriate.

L24, the impact of baseline is clear but this does not mean we need to change the baseline since the ecosystem may need time to adjust the changes in baseline.

**Reply:** This is exactly why we chose to compare the two baselines in the first place. Some parts of the marine ecosystem may still be adapted to the climate in the previous climatological average period (i.e., mid-1900s) and therefore experience MHW when compared to the 1961-1990 baseline, but not with the 1991-2020 baseline.

L29, Likewise, the removing of the linear trend does not make sense, since our scientists can remove the warming trend but the ecosystem cannot and it at least needs time to adjust the warming trend. It has not been unknown how long it will take for the ecosystem to get used to the new base line or warming trend.

**Reply:** The rationale here is that in some cases the slow and gradual climate change in itself is the biggest risk factor, whereas in other cases it is the instant shock of a short-lived anomaly that is the biggest risk factor. Removing the linear trend may reveal whether the shocks have become more severe. But this sentence is anyway referring to literature and previous studies, and we have now decided **not to** remove the linear trend when doing the MHW analysis.

L56, it may be better to say something why CTD data is used to your assessment.

**Reply:** We have changed "*assessing the performance of the two models*" with "*assessing the* **quality** *of the two models*" and added "***before we use the model results to calculate MHW statistics***" at the end of the sentence.

L83-84, this is not consistent with the statement of Oliver et al. (2018): "Note that when calculating the annual statistics of events which occur across several years, the duration and intensity are assigned to the start year of that event." Authors need to check and verify the consistency between the python code and the statement of Oliver et al. As the authors acknowledged that the frequency (maybe duration as well) may have been overestimated due to non-rational separation of MHWs across different years. More importantly, it is not clear whether MHWs are analyzed from the starting year (1991) to the ending year (2021). If yes, it is should be easy to fix the above problem. If not, I guess (based on the statement in the manuscript) the MHWs may have been analyzed every year from the January 1 to December 31. If this is the case, the MHWs may have been underestimated for those MHWs sustained from the end of year to the beginning of the next year. e.g. SSTs are above 90% over December 28-31 and January 1-4 of the next year, these SSTs may not be counted as an MHW if they are analyzed yearly, but should be counted as an MHW if they are analyzed for the entire period.

**Reply:** We have clarified this point by removing this confusing statement and rather stated that we have used the python algorithms by Oliver et al. (2018) using the default settings.

L98-100, it is easy to understand for the sea bottom MHWs if focuses are the ecosystem of the ocean bottom such as coral reefs. But this should be described much earlier in Introduction section and the Abstract.

**Reply:** We have added the following sentence in the Introduction to motivate the inclusion of bottom MHWs in the study:

*The Barents Sea is home to several important, commercial fish stocks,* ***both pelagic (e.g., capelin (****Mallotus villosus****) and Norwegian spring spawning herring (****Clupea harengus****)) and demersal***

*(e.g., northeast Arctic cod (Gadus morhua) and haddock (Melanogrammus aeglefinus)), in addition to a diverse marine ecosystem including large groups of marine mammals and sea birds as well as unique benthos communities.*

L103, should "Table 1" be Table 2? Descriptions are needed for Table 2.
**Reply:** Yes, we have corrected the text accordingly.

L111-115, it is not clear how these MHWs were diagnosed. Is it different from those based on Hobday et al. (2016) starting from L115?
**Reply:** The start of the sentence starting on L115 (referring to method by Hobday et al., 2016) has been deleted, because this information was unnecessary and only led to confusion (see also reply to reviewer #2).

Fig. 2., it is not very clear why the time series are from 2015 to 2018, as an example? How about the period from 1991-2014? Why the example of 2015-2018 was selected, and what are the implication for these MHWs. E.g. the connections from the 2015-16 El Niño event.
**Reply:** We believe the reviewer is referring to Fig. 3, which shows the duration and intensity of the MHW in 2016 in the four different regions. We have added the following to the first sentence leading up to Fig. 3:
*"To look for regional differences, we **chose to** investigate the 2016 MHW event, **which was the most severe MHW event detected in the Barents Sea as a whole,** in the four sub-regions depicted in figure 1"*

L129-130, since the negative trends of the bottom MHWs were not statistically significant, it might be safe to say "no significant trends were detected".
**Reply:** Changed as suggested

L168-169, the heat is not directly related to the influx, but to the convergence of influx and outflux. What is the change of the outflux from Barents Sea to the Arctic?
**Reply:** While we agree that the heat content of a volume is determined by the heat convergence within that volume, for the Barents Sea we may still assume that the heat content to a large degree is determined by the inflow from the southwest. This is because the outflow of oceanic heat to the northeast is almost negligible (about one order of magnitude smaller than the inflow) and the largest heat sink is heat lost to the atmosphere while the water is traversing the Barents Sea. It has been shown in several studies that the heat transport **from** the Barents Sea is small and that most of the oceanic heat is lost to the atmosphere in the Barents Sea (e.g., Gammelsrød et al., 2009; Lien & Trofimov, 2013; Smedsrud et al., 2013; Skagseth et al., 2020) and that increased inflow to the Barents Sea cause increased heat within the Barents Sea and reduced sea-ice cover (e.g., Onarheim et al., 2015; Lien et al., 2017).
We have added the following sentence to better substantiate our conclusions regarding the link between inflow and MHW events (see also reply to earlier comment):
*While we did not perform a closed heat budget calculation, we note that the oceanic heat carried by the downstream outflow from the Barents Sea has previously been reported to be smaller*

*than the inflow by an order of magnitude (Gammelsrød et al., 2009; Smedsrud et al., 2013). Furthermore, a previous study found that increased heat advection to the Barents Sea lead to increased ocean heat content in the interior Barents Sea (Lien et al., 2017).*

L172, What is the "turbulent heat", is it sensible, latent heat fluxes. How about solar radiation fluxes?
**Reply:** We have clarified this point by adding "*... the turbulent **(latent and sensible)** heat loss ...*" at both occasions where turbulent heat loss is mentioned.
Solar radiation fluxes are negligible during the DJF/winter period, due to the Polar Night at the latitude of the Barents Sea. We have also added this information explicitly.

Fig. 5., Can the heat flux analyses be applied to both surface and bottom MHWs? What results the differences between surface and bottom MHWs?
**Reply:** The surface heat fluxes affect the bottom MHW indirectly through vertical mixing during winter, while during summer the surface and bottom layers are usually separated by stratification. We have added the following sentence:
*Moreover, wind-driven mixing breaks down the upper water column stratification during winter, connecting the surface with the deeper layers.*

Table 3, "Number of marine heatwave events per year during the period 1961-2020" is very confusing and out of context. I think the same period of 1991-2021 should be analyzed and compared with different baseline periods of 1961-1990 and 1991-2020, which can also be compared with the results presented in section 3.1.
**Reply:** We have changed the title of the table to:
*"**Average frequency** of marine heatwaves +/- the decadal trend for two different baseline periods, 1961-1990 and 1991-2020."*

Tables 3-5, I assume these are for the surface MHWs, what about the bottom MHWs?
**Reply:** Tables 3-5 show results both for the surface and the bottom. We have now stated clearly in the first sentence of the paragraph that the calculations are done both for the surface and the bottom.

L237-238, Does this imply that the ice may be melted at the bottom while remained at the surface?
**Reply:** The sea ice affects the bottom only indirectly through the sinking of cold, brine-enriched water as a consequence of sea-ice formation at the surface. For clarification, we have added the following:
*"[...] sea-ice formation on nearby banks **and thus a reduction in the sinking of brine-enriched surface water**."*

L238, is this "sea-ice cover" the surface ice or bottom ice cover?
**Reply:** There is no sea ice on the bottom. See also the reply to the comment above.

**Reply to Reviewer #2**

This is a study of the characteristics of marine heatwaves in the Barents Sea. It aims to investigate the occurrence of surface and bottom heatwaves in four different environments in the Barents Sea, to explore the differences in marine heatwaves characteristics when changing baselines, and to focus on the most severe MHW events in their study area.

Overall, I vacillated between rejecting the paper and accepting it with major comments and decided to give the authors another chance to improve their work. I have major and minor comments, such as the following:

**Reply:** We would like to thank the reviewer for giving us a second chance and providing thoughtful comments and suggestions which helped improve the manuscript.

Also, please note that, in addition to addressing the issues raised by this reviewer, we have also updated all results, figures, and tables to also include the year 2022, because that year was not available at the time of the submission of the manuscript.

**Major comments:**

The language used to describe the results of this work is very general and has nothing to do with the MHW topic. To be more clear, there are some terminologies that are known to be used by researchers of different scientific topics, here in this paper they used different terminologies that are far from what should be used in a study like the MHWs, and this led to a lot of confusion when I read their paper. I doubt that the authors used some sort of paraphrasing software that caused these repeated mistakes in their manuscript. Here are examples of the mistakes to which I refer:

**Line 9:** It was written "Determining the amplitudes and extent of marine heatwaves". For MHWs, the term "**intensity**" was used to describe the level that the SST reached above the threshold climatology, while the term "amplitude" is usually used for ocean waves, sound waves, etc., not for MHWs. Also, the extent of the MHWs could be a vertical or spatial extent, so please specify which one you are referring to, and I think you mean vertical extent since you don't have any spatial maps in your paper.

**Reply:** Thank you for pointing out this lack of appropriate terminology. We have changed the terminology throughout the manuscript to align it with the standard terminology of other MHW studies.

**Lines 13 – 14:** The authors used the word "expression" in two different places and this word does not mean the same thing in the 2 places. The first time it said "to explore the frequency, intensity, and duration of marine heatwaves in the Barents Sea, as well as 13 their regional expression", and I suspect they were referring to the **regional occurrence** of MHWs. The second place was, "We find that major marine heatwaves are rather coherent throughout the region and have comparable surface and bottom expressions", and I believe using words like **occurrences or events** is more representative in this case. Please change it through the entire manuscript.

**Reply:** We have changed the terminology and made the text clearer and more coherent throughout the manuscript.

**Line 23:** "temperatures above the 90th percentile based on a fixed baseline", Hobday et.al., 2016 define an MHW when the SST exceeds either **the 90$^{th}$ percentile of the moving climatological average** or **a fixed threshold**, so the above statement is not correct, please correct it. Also superscript the "th" that is after 90.

**Reply:** We have revised the text to be consistent with the method we have applied.

**Line 24:** "Climate normal", it is usually referred to as climatological average or climatological mean, please change it.

**Reply:** Thank you for pointing out this. We have changed to ***climatological average*** throughout the manuscript.

**Line 32:** "When MHW are calculated for a whole region, regional heterogeneities will be lacking, thereby reducing the applicability of such an index" ??? which index???????

**Reply:** We have rephrased the sentence to clarify our point: "*When MHW are calculated **as a timeseries** for a whole region, regional heterogeneities may be masked, thereby reducing the applicability of **using the timeseries as** an MHW index.*"

**In the introduction:** you refer to other studies in the literature (e.g., line 37 (see e.g., Jakobsen and Ozhigin (2011) for a comprehensive overview)). Instead, try to make your introduction richer and like a story that gives the reader an idea of the topic you are studying. There are so many studies on MHWs and one of them deals with the Barents Sea (https://doi.org/10.3389/fmars.2022.821646). So you can use their work to introduce yours, and the authors should also mention how their work differs from the previous one. Also, you have a very unique area of study. I would have expected you to describe it in more detail in your introduction. This will help the reader to know more about the circulation in the Barents Sea, the physical properties and the atmospheric conditions, so that later when you talk about it, they can compare the normal and the extreme conditions.

**Reply:** Thank you for bringing this up. We agree that the introduction, as it reads now, require some background knowledge about the region. While part of the motivation for that was to keep the manuscript short, we have expanded the introduction to better familiarise the reader with the area.

**Lines 70 - 73:** "We choose these periods to examine the effect on MHW statistics of using different baseline periods. The first period, 1961-1990, was a relatively cold period in the Barents Sea region, whereas the period 1991-2020 has been relatively warm (e.g., González-Pola et al., 2020)". I don't really see any logic in comparing the MHWs defined by these two baselines, because the result is easy to expect. It was a cold period from 1961 to 1990, so this baseline will result in longer and more intense MHWs compared to the warm period baseline (1991-2020). If the author wants to study the results of MHWs using different baselines, I suggest you try to take a longer or shorter period for climatology, e.g. 25 or 35 years, but within the same period (warm or cold) so that you can check whether using less or more than 30 years of climatology could be reflected in the characteristics of MHWs or not. This will be very helpful for the studies examining the composite

events of the MHWs and, for example, low chlorophyll-a events, since the available data for chlorophyll-a start from 1998.

**Reply:** While the results of our comparison of the two baselines are predictable, our motivation was to demonstrate the aspect that different parts of the ecosystem have different adaptability to climate variability and change (see also reply to reviewer #1). However, we have performed new analysis as suggested, using different climatological average lengths (20 and 25 years as compared to the 30-year baseline analysis).

**Lines 101-102:** "Here, we compare modelled and observed near-bottom temperatures averaged in time (monthly) and space (see sub-regions, Fig. 1)." How did you manage to define the MHWs using the monthly data? According to the definition you use, it is a daily phenomenon.

**Reply:** Here we refer to the model evaluation with the aim of assessing the model's ability to represent the temperature variability seen in observations. To clarify that we are referring to the model evaluation and not the MWH analysis, we have changed "*Here*" to "***In this model quality assessment***".

**Lines 111 – 114:** "We first estimate MHW statistics based on the TOPAZ reanalysis for the full Barents Sea region (see Fig. 1 for area definition). Two distinct MHW events are identified in both the surface and bottom temperature time series. While the strongest MHW, in terms of cumulative effect (degree days), appeared in 2016 both near the surface and near the bottom, the second strongest MHW appeared in 2013 near the surface and in 2012 near the bottom".

Does it make sense that you find only 2 MHW events over 31 years ( 1991 – 2021)? Especially the other study (Mohammed et.al., 2022) stated that over the 39 years (1982- 2020) In total, 72 MHWs were recorded in the whole Barents Sea over the 39-years study period with a total of 1,068 MHW days.

**Reply:** There are more than two MHW events identified, but there are two events that clearly stand out among all the events. We have rewritten this part starting with summarizing how many events were detected and their average characteristics, and then we go on to describe the two most severe events because the two really stands out among all the other events.

How is it possible to find MHWs on the bottom (2012) that cannot be seen on the surface? Especially your area is very shallow (maximum depth about 400 m), where does the heat come from?

**Reply:** While the investigation of possible mechanisms behind this finding is beyond the scope of such a short paper documenting MHWs, we have mentioned some possible mechanisms based on published literature on anomaly propagation and development in the Barents Sea:

*"While an investigation of possible mechanisms for this decoupling between the surface and the bottom is beyond the scope of this work, we note that the 2012/13 MHW event was preceded by an extraordinarily large temperature anomaly but close to average volume transport in the Atlantic Water entering the Barents Sea to the southwest (e.g., ICES, 2022), as opposed to extraordinarily large volume transports preceding the 2016 MHW event (see below for more details). Moreover, previous studies have suggested that temperature anomalies that are advected into the Barents Sea at depth during the stratified summer season, can reemerge at the*

*surface further downstream through vertical mixing during the following winter (e.g., Schlichtholz, 2019)."*

**Line 115:** "When applying the MHW definition provided by Hobday et al. (2016)" . So what was the method used to define MHWs in lines 111 – 114?
**Reply:** We have deleted this part of the sentence because it provided no necessary information and only acted to provide confusion.

I would like to ask you if you have studied the role of the sea ice cover cycle in the region for the occurrence of MHWs?
**Reply:** No, we have not done that. While we agree that it might be of interest, we find such additional analysis to be beyond the scope of this manuscript given the length limitations for contributions to the Ocean State Report. But we do indicate that sea ice may play an important role, as seen in the results in the Northeast Basin (which is seasonally ice covered).

**Minor comments:**
In your data section please specify the periods that you used for each dataset
**Reply:** We have added information about the periods covered by the different datasets.

Lines 151 - 152 rephrase it to be "In the Bear Island Trench and Pechora Sea, the maximum intensities of MHW near the surface are about twice the maximum intensities near the bottom".
**Reply:** Done

The figures: Put labels (A, B, ….etc) on the panels so it easier to be easier to be interpreted.
**Reply:** Labels have been added to the figures as suggested.